

# Exploitation of the far-offshore wind energy resource by fleets of energy ships - Part 2: Updated ship design and cost of energy estimate

Aurélien Babarit[1,2], Félix Gorintin[2,3], Pierrick de Belizal[3], Antoine Neau[3], Giovanni Bordogna[4], Jean-Christophe Gilloteaux[1]

[1]LHEEA, Ecole Centrale de Nantes - CNRS, Nantes, 44300, France
[2]INNOSEA, Nantes, 44300, France
[3]Farwind Energy, Nantes, 44300, France
[4]Blue Wasp Marine, Rotterdam, 3035TA, The Netherlands

*Correspondence to*: Aurélien Babarit (aurelien.babarit@ec-nantes.fr)

**Abstract.** This paper deals with a new concept for the conversion of far-offshore wind energy into sustainable fuel. It relies on autonomous sailing energy ships and manned support tankers. Energy ships are wind-propelled ships that generate electricity using water turbines attached underneath their hull. Since energy ships are not grid-connected, they include onboard power-to-X plants for storage of the produced energy. In the present work, the energy vector X is methanol.

In the first part of this study (Babarit et al., 2020), an energy ship design has been proposed and its energy performance has been assessed. In this second part, the aim is to estimate the energy and economic performance of such system.

In collaboration with ocean engineering, marine renewable energy and wind-assisted propulsion's experts, the energy ship design of the first part has been revised and updated. Based on this new design, a complete FARWIND energy system is proposed, and its costs (CAPEX and OPEX) are estimated. Results of the models show (i) that this FARWIND system could produce approximately 70,000 tonnes of methanol per annum (approximately 400 GWh per annum of chemical energy) at a cost in the range 1.2 to 3.6 €/kg, (ii) that this cost may be comparable to that of methanol produced by offshore wind farms in the long term, and (iii) that FARWIND-produced methanol (and offshore wind farms-produced methanol) could compete with gasoline on the EU transportation fuel market in the long term.

## 1 Introduction

To date, fuels such as oil, natural gas and coal account for approximately 80% of primary energy consumption globally (BP, 2018). Although this share is expected to decrease with the development of renewable power generation and the electrification of the global economy, some sectors may be difficult to electrify (e.g., aviation, freight). Therefore, if a global temperature change of less than 2°C—as set out in the Paris agreement—is to be achieved, there is a critical need to develop low-carbon alternatives to fossil fuels.



To address this challenge, we proposed in (Babarit et al., 2019) an energy system (FARWIND) which could convert the far-offshore wind energy resource into sustainable fuel using fleets of energy ships, see Fig. 1. Energy ships are ships propelled by the wind which generate electricity by means of water turbines attached underneath their hulls. The generated electricity is converted into fuel using onboard power-to-gas (PtG) or power-to-liquid (PtL) plants. In the proposed system,
the fuel is methanol (see Babarit et al., 2020 and Babarit et al., 2019 for a detailed explanation of the choice of methanol rather than carbon-free fuels like hydrogen or ammonia).

The produced methanol is collected by tankers which are also used to supply the energy ships with the necessary feedstock (carbon dioxide) for power-to-methanol conversion. Of course, the $CO_2$ supply source must be sustainable for that system to produce sustainable methanol. Therefore, the $CO_2$ must be captured either directly or indirectly from the
atmosphere. Possible options include direct air capture (Keith et al., 2018), $CO_2$ capture from flue gases from biomass or FARWIND-produced methanol combustion, and CO2 from biogas upgrading (Li et al., 2017; Irlam, 2017).

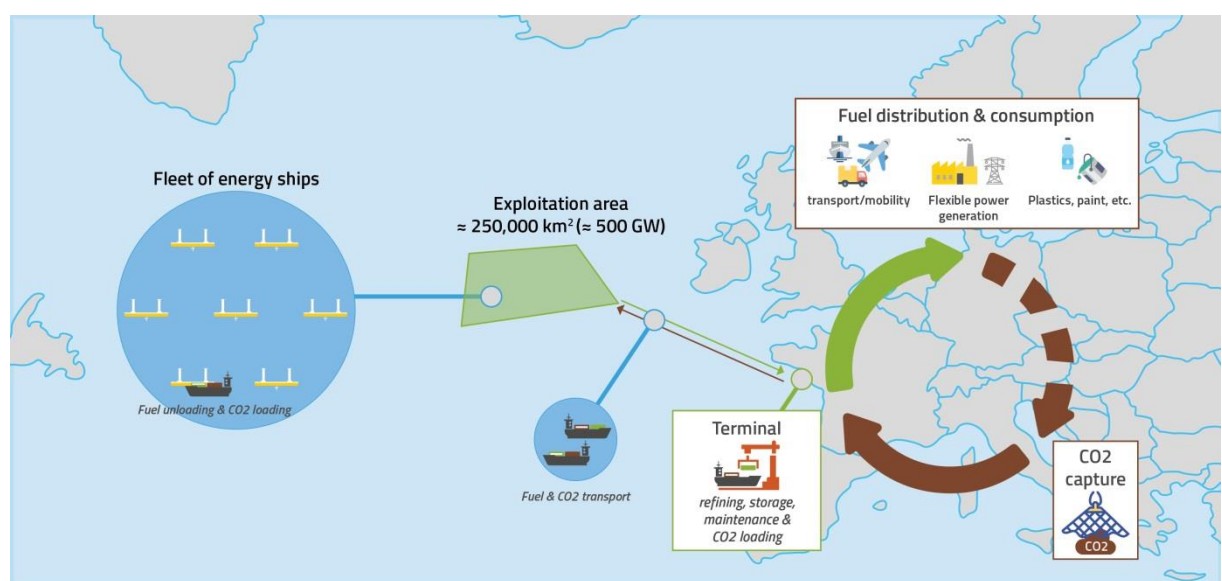

**Figure 1 The concept of sustainable methanol production from far-offshore wind energy by FARWIND energy systems.**

The overall aim of the present study is to investigate the energy and economic performance of the FARWIND energy
system. A preliminary energy ship design was proposed in (Babarit et al., 2020) and its energy performance was investigated. This design has been reviewed by ocean engineering and marine renewable energy's experts of the Marine Energy Alliance European project (EMEC, 2020); and wind-assisted propulsion experts (Blue WASP, 2020). Based on their feedback, the ship design has been updated; and an economic model has been developed. The aim of the present paper is to present that updated design, the economic model, and the resulting cost of energy.

The remainder of this paper is organized as follows. In section 2, the specifications of the updated design and its velocity and power performance are presented. In section 3, the specifications of the proposed energy system are presented, and its annual methanol production is estimated. Estimates of expenditures for the proposed energy system are provided and



discussed in section 4. Using those estimates and the estimates of annual methanol production, the cost of energy is estimated in section 5 and market perspectives are discussed. Conclusions are presented in Section 6.

## 2 Specifications of the updated energy ship design


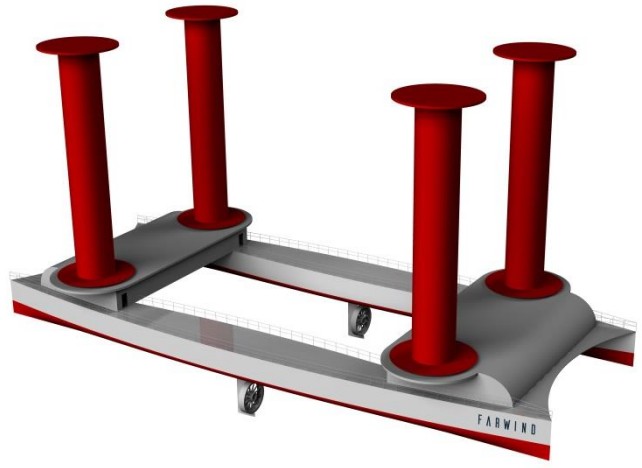

**Figure 2  Artist's view of the proposed energy ship design.**

The energy ship design considered in this study is a revision of that presented in (Babarit et al., 2020), see Fig. 3. It is still an 80 m long catamaran with four 5 m diameter Flettner rotors and two water turbines. The hull shape is the same.

However, the height of the Flettner rotors is increased from 30 m to 35 m, and the rated power of each water turbine is reduced from 900 kW to 800 kW. The complete characteristics of the ship are summarized in Tab. 1. Justifications for the data shown in the table are provided in the following sections.

|  | Unit | Value |
|---|---|---|
| Hull |  |  |
| Length | m | 80 |
| Breadth | m | 31.7 |
| Draught | m | 1.6 |
| Displacement | t | 1,035 |
| Structural mass | t | 560 |
| Wind propulsion |  |  |
| Type | - | Flettner rotors |
| Number | - | 4 |
| Rotor height | m | 35 |
| Rotor diameter | m | 5 |
| Rotor mass | t | 79 |
| Rotor drive power (max) | kW | 143 |



| Water turbine | | |
|---|---|---|
| Number | - | 2 |
| Turbine diameter | m | 4 |
| Rotor-to-electricity efficiency ($\eta_3$) | - | 75% |
| Turbine mass | t | 15 |
| Rated power | kW | 800 |
| Auxiliaries subsystems | | |
| Power consumption | kW | 50 |
| Auxiliaries subsystems mass | t | 41 |
| Power-to-methanol plant | | |
| Electrolyzer rated power | kW | 1,130 |
| Electrolyzer mass | t | 28 |
| Desalination unit rated power | kW | Negligible |
| Desalination unit mass | t | Negligible |
| $H_2tMeOH$ plant capacity | kg/h | 100 |
| H2tMeOH plant mass | t | 17 |
| Storage tanks | | |
| $CO_2$ storage capacity | t | 23 |
| CO2 storage tank mass (empty) | t | 15 |
| Methanol storage capacity | t | 17 |
| Storage tank mass | t | 4 |

**Table 1 Specifications of the updated energy ship design**

## 2.1 Rotors

The rotors technical specifications (dimensions, mass, maximum rotor drive power) used in this study are based on that of the largest currently available Flettner rotor (Norsepower, 2021).

The propulsive force of a Flettner rotor depends on the lift coefficient $C_L$ and the drag coefficient $C_D$ (Equation 2 in (Babarit et al., 2020), which depend themselves on the ratio of the rotational velocity of the rotor to the apparent wind speed (spin ratio $SR$). In (Babarit et al., 2020), we used the experimental data of (Charrier, 1979) for the aerodynamic coefficients

of a Flettner's rotor as function of the rotor's spin ratio $SR$. However, these experiments were carried out at low Reynolds numbers (~10,000). Recently, formula based on full scale data (Reynolds number over $10^6$) have been published (Tillig & Ringsberg, 2020). That data has been used in the present study (Fig. Figure 3) as it corresponds better to real conditions.

Moreover, rotors must be powered for them to spin. In (Babarit et al., 2020), we assumed that the rotors power consumption is constant (four times 40 kW), whereas in practice it depends on the wind loading. In their work, (Tillig &





Ringsberg, 2020) have developed a formula to estimate a rotor's power consumption as function of the spin ratio. We used that formula in the present study.

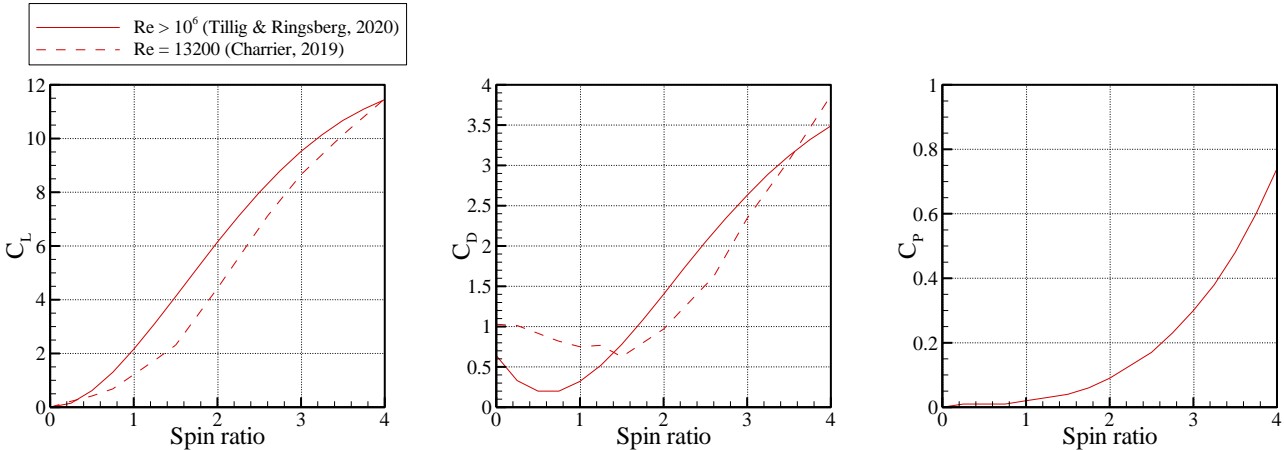

**Figure 3 Comparison of aerodynamic coefficients of Flettner rotors according to (Charrier, 1979) and (Tillig & Ringsberg, 2020)**

    In (Babarit et al., 2020), the effect of aerodynamic interactions between rotors was neglected. In the present study, it has
been estimated using the approach proposed by (Roncin & Kobus, 2004) in which each rotor is modelled by a horseshoe vortex. The implementation follows that of (Bordogna, 2020).

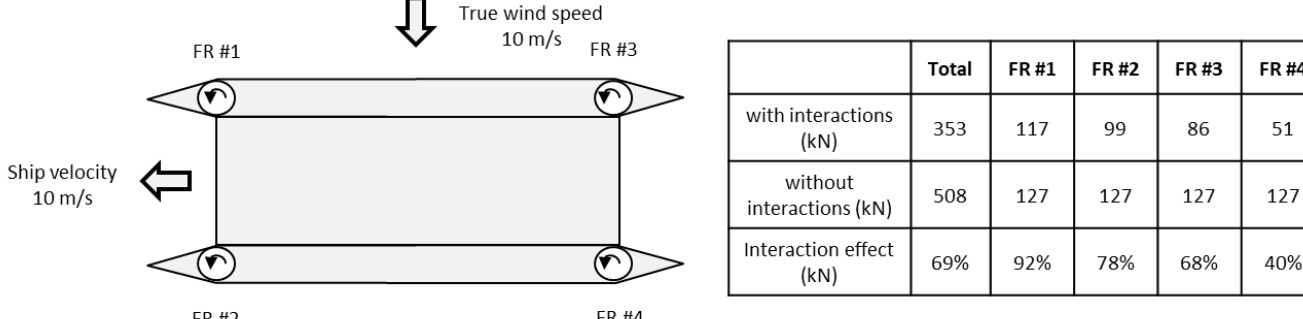

| | Total | FR #1 | FR #2 | FR #3 | FR #4 |
|---|---|---|---|---|---|
| with interactions (kN) | 353 | 117 | 99 | 86 | 51 |
| without interactions (kN) | 508 | 127 | 127 | 127 | 127 |
| Interaction effect (kN) | 69% | 92% | 78% | 68% | 40% |

**Figure 4 Effect of aerodynamic interactions on the propulsive force**

    The total propulsive force (with and without aerodynamic interactions) and the propulsive force from each rotor are
shown in Figure 4 for rated conditions (10 m/s true wind speed, 90° true wind direction, $SR = 3$, 10 m/s ship velocity). They show that the interaction effect cannot be neglected as the total propulsive force is 69% of that without interactions. A similar interaction effect has been found for other wind speeds (not reported here). Consequently, the model has been updated. The total propulsive force (Eq. 2 in (Babarit et al., 2020.) has been reduced by a constant factor of 30% for all wind conditions.





The Earth atmospheric boundary layer was also not considered in the energy performance estimate in (Babarit et al., 2020). In the present study, a power law has been assumed with an exponent of 0.14. Thus, in the updated model, the wind speed $W$ in Eq. 3 of (Babarit et al., 2020) is given by:

$$W = W_{10}\left(\frac{Z}{10}\right)^{0.14}$$

(1)

Where $Z$ is 22.5 m (half the height of the rotor + 5 m).

## 2.2 Hull

The hull shape is the same as for the preliminary design. However, the hull mass estimate has been refined. The revised mass estimate is based on a preliminary scantling of the hull structure which has been developed using rule NR600 of Bureau Veritas (EMEC, 2020). The corresponding hull weight estimate is 560 t, which is more than twice the estimate of

the preliminary design. Moreover, the updated design assumes taller rotors (35 m), which are 20 tons heavier than the 30 m rotors of the preliminary design. Consequently, the total displacement of the updated design is 1,035 tons (660 tons for the preliminary design).

Due to the increased displacement, the wetted surface increases to 1,064m². The wave resistance coefficient has also been updated (see Figure 5). As for the preliminary design, it was calculated using the software REVA (Delhommeau

and Maisonneuve, 1987).  One can see that the residuary resistance coefficient (wave making) is greater for the updated design than for the initial design, which is due to the increased displacement.

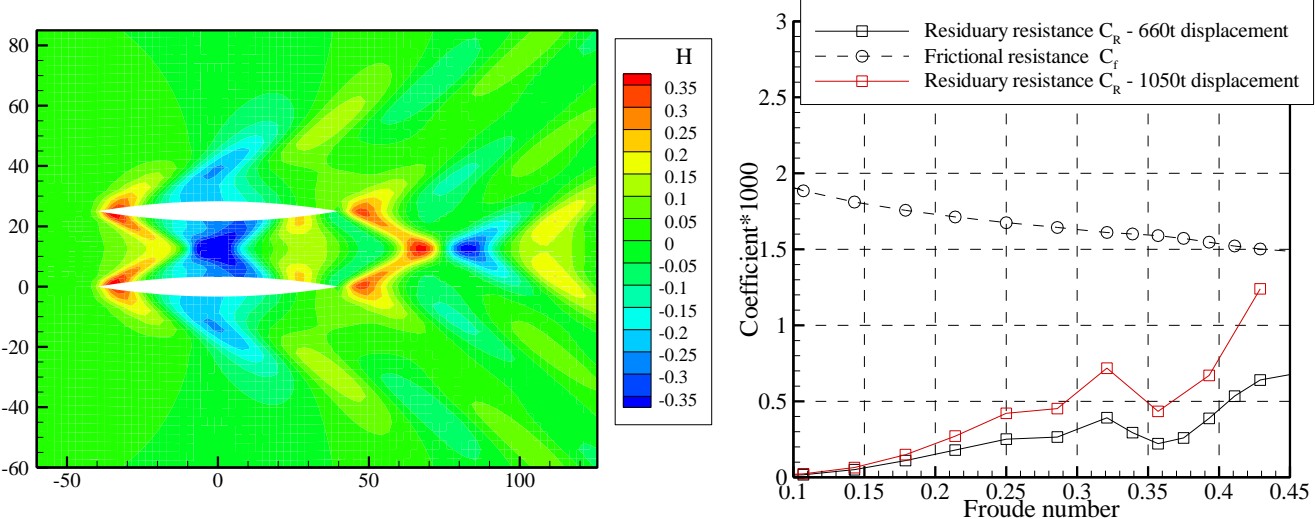

**Figure 5  Left: wave pattern around the ship hull of the updated design for Froude number = 0.357 (10 m/s ship velocity). Right: hydrodynamic coefficients of the initial and updated design.**



### 2.3 Water turbine

The water turbines' dimensions are the same as for the initial design (4 m diameter rotor). However, their mass is increased to 15 tons each (7.4 tons each for the initial design). Based on expert's advice, the water turbine's energy efficiency has been reduced to 75% (80% for the initial design). The rated power is decreased to 800 kW (900 kW for the initial design).

### 2.4 Power-to-methanol plant

For rated wind conditions (10 m/s true wind speed, 90° true wind angle), the ship velocity is almost 10 m/s (see section 2.7). The water turbines' power production is 1,600 kW. The Flettner rotors' power consumption is approximately 420 kW. Assuming a further 50 kW power consumption for the auxiliary subsystems, the net power production available to the electrolyzer of the power-to-methanol plant is 1,130 kW (1,420 kW for the initial design). The weight estimate of an electrolyzer of such rated power is 28 t (35 t for the initial design).

Assuming the same 60% efficiency for the electrolyzer as for the initial design, the rated power of the H2-to-methanol plant is 680 kW (850 kW for the initial design). Its weight estimate is 17 t (24 t for the initial design).

### 2.5 Storage tanks

The capacities of the storage tanks ($CO_2$ and methanol) are set such as they can accommodate 7 days of production at rated power (approx. 17 t of methanol). Thus, the $CO_2$ tank weight is 15 t and that of the methanol tank is 4 t.

### 2.6 Auxiliary equipment

As for the initial design, the weight of the auxiliary subsystems is taken equal to 10% of the total mass budget excluding the hull weight (41 t).





## 2.7 Power production charts

**Figure 6** Ship velocity (a), power generation (b), rotors power consumption (c) and net power (d) of the updated ship design as function of wind conditions. TWA stands for true wind angle.

130



The velocity and power performance of the updated design has been calculated using the model presented in (Babarit et
al., 2020). The results are shown in Figure 6 as function of the wind conditions (true wind speed and true wind angle).

Overall, the velocity and power performance of the updated design resemble that of the initial design (albeit 10 to 20%
smaller). As for the initial design, rated power (1,600 kW) is achieved from a true wind speed of 10 m/s and a true wind
angle of 90°. However, a major difference is that the rotors power consumption depends on the spin ratio in the updated
design velocity and power performance prediction model, whereas it was fixed in (Babarit et al., 2020). Therefore, the net
power keeps increasing with increasing wind speed (see panel (d)) despite the generated power has reached rated power
(1,600 kW).

As for the initial design, the water turbine's induction factor and the rotors' spin ratio were optimized to maximize
power production for each data point while satisfying constraints (maximum rotation velocity and thrust force for the rotors,
maximum power generation for the water turbine). Due to those constraints, there can be several settings (induction factor,
spin ratio) for the same power generation, which explain the noisy behavior for the ship velocity in panel (a).

## 3 Specifications of the proposed FARWIND energy system

In the FARWIND energy system concept, the energy ships are deployed in fleets and are supported by tankers
which collect the produced methanol and transport it to a shore-based terminal, see Figure 1. The tankers also provide the
energy ships with $CO_2$. In this section, we estimate the characteristics and number of the tankers, and the number of energy
ships in a FARWIND system.

### 3.1 Tanker design

In the considered energy ship design, the methanol storage tank capacity allows storage of one week of methanol
production at full capacity. Therefore, each energy ship of the fleet must meet a tanker for methanol collection and $CO_2$ refill
at least once a week (to avoid stops in the production process because the methanol tank is full or because the $CO_2$ tank is
empty).

Thus, let us estimate the number of energy ships that can be served by one tanker. This depends on the duration of
the $CO_2$-loading and methanol-unloading operations. We assume that these operations take six hours on average, and that
they are carried out continuously (including at night). Therefore, one tanker can service 28 energy ships per week (7
days/week x 24 hours/day / 6 hours/operation). As the capacity of an energy ship's methanol tank is 17 tonnes (23 tonnes for
the $CO_2$ tank), the tanker may collect up to 473 t of methanol and supply 650 t of $CO_2$ every week.

It is assumed that the tankers are operated by a crew, and that the duration of their mission is four weeks. At the end
of each four-weeks mission, the tanker returns to a shore-based terminal to change crew, unload the methanol, and load $CO_2$.





Therefore, their total methanol capacity must be 1,891 t (4 weeks x 473 t/week) and their total $CO_2$ capacity must be 2,601 t (4 w x 650 t/w). Assuming the $CO_2$ will be stored as liquid in a cryogenic storage tank, and extrapolating from (Chart, 2019),

the empty weight of a 2,600 t capacity $CO_2$ storage vessel is estimated to be 1,700 t. For methanol, the mass of the required tank is estimated to be 410 t. The tanker will be carrying maximum cargo weight (4,720 t) when it leaves the terminal (full $CO_2$ tank and empty methanol tank). This cargo weight is relatively similar to the average vessel size of small crude oil (3,600 deadweight (dwt)), chemical (4,900 dwt) and LPG vessels (3,500 dwt) (Lindstad et al., 2012). According to (MAN Energy Solutions, 2019), the propulsion power of a 5,000 t deadweight bulk carrier is 1,410 kW for a service speed of 12

knots. These are the values which we used for the service speed and propulsion power of the tanker.

### 3.2 FARWIND system design

Following (Babarit et al., 2018), it is assumed that the fleet of energy ships is deployed at a distance of 1,000 km from the terminal. Therefore, the tankers must travel 1,000 km to meet the energy ships, and a further 1,000 km when returning to the terminal. At a service speed of 12 knots, the tanker's round-trip will take 90 hours. Considering the duration of

unloading/loading operations and other maintenance operations, we estimate that the tanker will be away from the fleet of energy ships for a duration of one week.

To ensure continuous operation of the energy ships, the tanker must be replaced immediately when it leaves the production zone. Therefore, each group of 28 energy ships must be supported by more than one tanker. It can be shown that the minimum number of tankers per fleet must be at least 1.25, meaning that the optimal FARWIND system comprises a fleet of

112 energy ships supported by five tankers. Over a year, the number of roundtrips between the terminal and the production zone is 10.4 for each tanker. The maximum methanol production of that system (assuming 100% capacity factor for the energy ships) is approximately 100,000 t per annum.

### 3.3 Annual methanol production of the proposed FARWIND system design

Since energy ships are mobile, their route schedules can be dynamically optimized based on weather forecasts in order to

maximize energy production. This was performed using a modified version of the weather-routing software QTVLM (Abd-Jamil et al., 2019). The coordinates of the starting and arrival point are: N 50.5; W 18.9 (approximately 1,000 km from the port of Brest, France). Over the three years 2015, 2016 and 2017, it was found that an average capacity factor of over 75% can be achieved.

That estimate does not consider downtime due to maintenance (availability). According to (Sheng, 2013) and (Pfaffel, 2017),

the failure rate of wind turbines is in the order of one failure per annum. Given the greater complexity of the energy ship system (additional energy conversion subsystems in comparison to a wind turbine e.g. power-to-methanol plant), it is assumed that the average failure rate of energy ships is two failures per annum. The corresponding downtime is driven by accessibility and repair time. As accessibility at sea can be challenging and as energy ships are mobile, it is assumed that




most of the repairs are performed at a port. Moreover, it is assumed that despite the failure, the energy ship is able to sail to
that port at an average velocity of 10 knots (corresponding to half the rated velocity) without assistance (e.g. tug boat).
Assuming that the distance between the production area and a port (with a dedicated shipyard) is 1,000 kms, it would take
approximately two days for that energy ship to go to the port. Assuming a further 3 days for the repair and 2 days for the
energy ship to go back to the production area, the downtime per failure is 7 days. Thus, for a failure rate of two failures per
annum, the total downtime per annum is two weeks corresponding to a 96% availability.

Taking into account that availability estimate, it appears that a capacity factor of 72% can be achieved. The corresponding
annual methanol production would be 70,600 t per annum. Note that it would require the supply of 97,400 t of $CO_2$, as the
production of 1 kg of methanol requires 1.38 kg of CO2.

## 4 Estimation of expenditures

### 4.1 Capital cost of a first of a kind energy ship

|  | Cost (k€) |
|---|---|
| **Energy ship** |  |
| Hull | 1,100 – 2,200 k€ |
| Flettner rotors | 4,200 – 4,900 k€ |
| Water turbines | 1,300 – 2,700 k€ |
| Auxiliaries, assembly and systems integration | 1,300 – 2,000 k€ |
| Electrolyzer | 1,100 – 2,200 k€ |
| $H_2$-to-methanol plant | 400 – 700 k€ |
| Fresh water production unit | Negligible |
| Liquid $CO_2$ tank | Negligible |
| Methanol tank | Negligible |
| Power-to-methanol plant indirect cost (installation and assembly, etc.) | 300 – 2,900 k€ |
| **Total** | **9,300 – 16,700 k€** |

**Table 2  Estimates of the capital cost of a prototype of the proposed energy ship**

Tab. 2 shows estimates of the capital cost of a prototype of the proposed energy ship.

The hull cost estimate was obtained using the usual approach which is to multiply the hull weight by a price per ton of
fabricated steel. That price includes procurement and workforce required for hull construction. Thus, it depends on steel
market price and shipyard location. The typical cost range is 2,000 €/t (South-East Asia construction) to 4,000 €/t (Northern
Europe). The hull weight estimate being 560 t, we retain a hull cost in the range 1,100 to 2,200 k€.





According to (Kuuskoski, 2019), the cost of four 30 m Flettner rotors is in the range 3,000 to 3,500 k€. For four 45 m tall Flettner rotors, we assumed that the cost is approximately proportional to the rotor mass excluding foundation. That mass being 42 tonnes for a 30 m tall rotor and 59 tonnes for a 35 m tall rotor (Norsepower, 2021), we retain a Flettner rotors' cost in the range 4,200 to 4,900 k€.

The water turbine cost estimate assumes that it is proportional to its rated power. We assume that the price is in the range 800 to 1,700 €/kW, which yields a water turbine cost in the range 1,280 to 1,720 k€.

Ship common systems, ship assembly and systems integration typically represent 20% of the total cost of a ship according to (Shetelig, 2013). We applied this ratio to the sum of the hull cost, Flettner rotor cost and water turbines cost. The other equipments were not taken into account because their installation factor is taken into account separately.

Holl et al. (Holl et al., 2016) has developed scaling laws for the cost of the electrolyzer and the freshwater production unit based on market surveys. They depend on the nominal power of the equipment. Applying the electrolyzer scaling law to the 1,130 kW capacity electrolyzer of the energy ship results in an estimated cost of 1,400 k€, equivalent to 1,250 €/kW. This is in agreement with the range 1,000 to 1,950 €/kW reported in (Schmidt et al., 2017) for PEM electrolyzers (which we used in this study). As for the freshwater production, the application of the scaling law of Holl et al. yielded a cost estimate of 9 k€,

which is very small in comparison to the other costs.

According to (Brynolf et al., 2018), the cost of a hydrogen-to-methanol plant is in the range 600 – 1,200 €/kW of methanol. As the estimated efficiency of the power-to-methanol conversion process is 49% (Babarit et al., 2020), it corresponds to 300 to 600 €/kW of electrolyzer input power. Thus, we retain 400 – 700 k€ for the hydrogen-to-methanol plant capital cost.

For the liquid $CO_2$ and methanol storage tanks, suppliers and prices can be found on the internet (e.g.

https://www.gitank.com/methanol-storage-tanks, (Chart, 2019)); typical costs are 300 €/ton of capacity for methanol and 1,000 €/ton of capacity for liquid $CO_2$. Overall, their costs are negligible in comparison to other costs.

The electrolyzer and hydrogen-to-methanol costs do not include installation and assembly, transportation, building, etc. Those costs are usually taken into account using an installation factor. According to (NREL, 2014), the lower end of the installation factor is 1.2 and up to 2 for the higher end. This leads to a cost of 300 – 2,900 k€.

**4.2 Capital cost of a first of a kind FARWIND energy system**

According to the discussion in section 3.2, a FARWIND energy system should include a fleet of 112 energy ships and 5 tankers. One can expect the unit cost for a fleet of 112 energy ships to be significantly smaller than the cost of an energy ship prototype. To take this into account, a learning rate of 10% was assumed on the unit cost of the energy ship as function of the built capacity, see Tab. 2. It can be noted that such learning rate corresponds to what was observed for wind turbines

(Lindman and Soderholm, 2012). It leads to a range of capital cost of 620 to 1,110 M€ for the first fleet of energy ships. It corresponds to an average unit cost of 5,500 to 9,900 k€ per energy ship.

For the tanker, according to (Lindstad et al., 2012), the price of commercial ships is in the range 500 € to 4,750 € per ton of dwt, depending on the type of ships and size. The lower price is for crude oil tankers greater than 140,000 dwt, while the



higher price is for roll-in/roll-off (ro-ro) ships of 7,000 dwt. In the present study, we retain a cost range of 2,500 to 4,000
€/ton of deadweight, leading to a tanker cost in the range 12,500 to 20,000 k€.

Thus, overall, the total capital cost of a FARWIND system comprised of 112 energy ships and 5 tankers is expected to be in the range of 680 to 1,210 M€ (3,700 to 6,700 k€ per megawatt of installed capacity).

### 4.3 Operational expenditures

Expected operation and maintenance (O&M) costs, including the cost of $CO_2$ supply, are summarized in Tab. 3 and detailed
below.

| | O&M cost (in % of capital cost of equipment per year) |
|---|---|
| **Energy ship** | |
| Hull | 2% |
| Flettner rotors | 3.5% |
| Water turbine | 4 - 13% |
| Auxiliaries | 2% |
| Electrolyzer | 7.5 – 11.5% |
| $H_2$-to-methanol plant | 9 - 13% |
| Fresh water production unit | 10 - 20% |
| Liquid $CO_2$ tank | 2% |
| Methanol tank | 2% |
| **Total** | **4.8 – 8.5%** |
| **Tanker** | 4 - 10% |
| **FARWIND system** | |
| Energy ships maintenance | 24 – 58 M€/y |
| Tankers O&M | 3 – 10 M€/y |
| $CO_2$ supply | 2 – 19 M€/y |
| Insurance cost | 4 – 15 M€/y |
| **Total (including $CO_2$ supply and insurance cost)** | **4.5 – 8.3%** |

Table 3 **Estimates of the operation and maintenance of a first-of-a-kind FARWIND energy system**

### 4.3.1 Energy ships and tankers operation and maintenance cost

According to (Holl et al., 2016), the maintenance cost of the water turbine is in the range 4 to 13% of the capital cost, and that of the freshwater production unit is between 10 and 20%. According to (Chardonnet et al., 2017), the maintenance cost
for the electrolyzer is in the order of 4% of capital cost. It is 2 -5% according to (Brynolf et al., 2018). It is unclear whether those maintenance takes into stack replacement. Indeed, PEM electrolyzers' stack lifetime is in the order of 50,000 hours.





Thus, assuming a capacity factor of 72%, they would have to be replaced every 8 years. According to (Brynholf et al., 2018), stack replacement cost is 60% of the electrolyzer cost. It leads to an additional 7.5% maintenance cost for the electrolyzer. Thus, we retain 7.5 – 11.5% for the maintenance cost of the electrolyzer. The same range is assumed for the hydrogen to
methanol plant.

For the Flettner rotors, the maintenance cost is expected to be in the order of 3.5% of the rotors' capital cost (Kuuskoksi, 2020). For the other subsystems (hull, auxiliaries, storage tanks), it is expected that the maintenance costs would be small; a 2% maintenance cost was arbitrarily selected. Overall maintenance costs for the energy ship are thus in the order of 3.7 to 5.3%.

For the tanker, following (Holl et al., 2016), we estimate operation and maintenance costs to be 4 to 10%.

### 4.3.2 CO₂ supply cost

The ambition of the FARWIND energy system is to provide a sustainable alternative to the use of liquid fossil fuels (e.g. oil). Therefore, as mentioned in the introduction, the $CO_2$ must be captured directly or indirectly from the atmosphere.

According to (Keith et al., 2018), the cost for direct air capture (DAC) using large-scale wet absorption DAC technology is
in the range 80 to 204 €/ton of $CO_2$. The cost of $CO_2$ capture from biogas upgrading is in the order of 15 to 100 €/ton of $CO_2$ (Li et al., 2017). In the case of $CO_2$ capture from flue gases from combustion of biomass or FARWIND-produced methanol, the cost of carbon capture is in the order of 35 to 50 €/ton (assuming that it would be similar to that for capture of $CO_2$ from power production processes involving coal or natural gas (Irlam, 2017)). Note that for both biogas upgrading and biomass or methanol combustion, the $CO_2$ concentration in the source is much greater than in ambient air, which results in a more
effective capture than with DAC.

Carbon dioxide may also be extracted from seawater (Willauer et al., 2012). Indeed, some of the $CO_2$ present in the atmosphere dissolves in the ocean. However, this new technology is in its early stages of development (Willauer et al., 2017).

In any case, the captured $CO_2$ must be liquefied for efficient transportation. The energy requirement for $CO_2$ liquefaction is
in the order of 0.1 kWh/kg$_{CO2}$ according to (Oi et al., 2016), which is low; hence its associated cost is expected to be negligible.

Therefore, we estimate the cost of $CO_2$ production to be in the range 20 to 200 €/ton. As 97,400 t of $CO_2$ are required to produce 70,600 t of methanol, the $CO_2$ supply cost is estimated to be in the range 2–20 M€ per annum.

### 4.3.3 Insurance cost

Insurance cost is generally taken as 0.6% of CAPEX per year for vessels at the concept stage. However, for a new technology, this percentage of CAPEX may be higher, potentially as high as 1 - 2%. In this study, we have retained 0.6 – 1.2%.





# 4 Cost of energy estimates

## 4.1 Short-term cost

The levelized cost of methanol *LCOM* can be calculated as (Holl et al., 2016):

$$LCOM = \frac{CRF + \lambda}{AMP} I$$

(1)

where $I$ is the total capital cost, $\lambda$ is the total O&M rate, $AMP$ is the annual methanol production, and $CRF = \frac{i(1+i)^n}{((1+i)^n - 1)}$ is the capital recovery factor, in which $i$ is the weighted average cost of capital (WACC) and $n$ is the lifetime in years. Assuming a

WACC in the range 6–10% and a lifetime of 20 - 25 years, the capital recovery factor is in the range 7.8–11.7%. The methanol cost is thus in the range 1.2–3.6 €/kg (225 to 660 €/MWh$_{th}$).

This cost is three to nine times greater than current market price for methanol (0.4 €/kg ≈ 72 €/MWh in the first quarter of 2021). However, it does not consider a price on GHG emissions. At least 0.675 kg of $CO_2$ is produced per kg of methanol produced using conventional processes (which are based on coal or natural gas) (Martin and Grossmann, 2017). In 2018, the

carbon tax was 44.6 €/ton in France and 110 €/ton in Sweden; if $CO_2$ emissions were taken into account, the methanol price would increase by 6 €/MWh$_{th}$ and 13 €/MWh$_{th}$ respectively. Thus, unfortunately, even with a rather significant carbon tax, the cost of methanol produced with a first-of-a-kind FARWIND system would not be competitive.

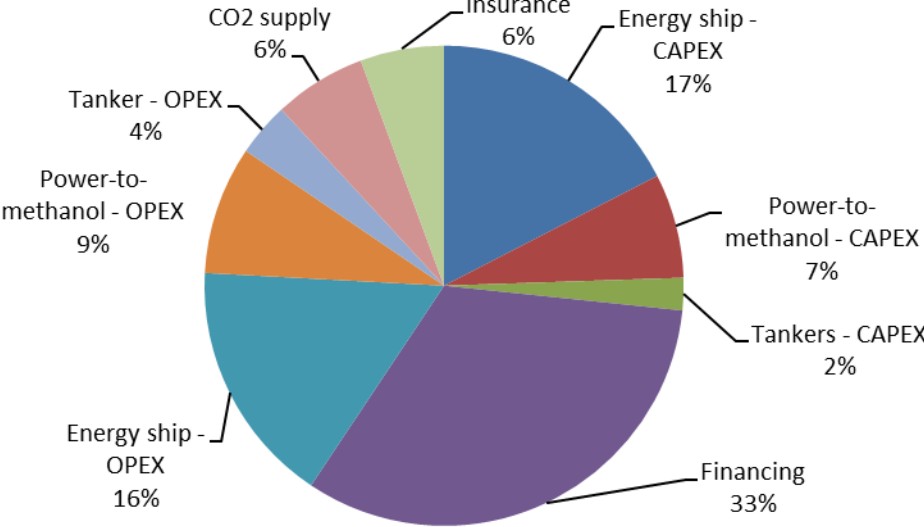

**Figure 7 Cost breakdown of methanol produced by a first-of-a-kind FARWIND system. The shown data corresponds to an**
**average cost scenario (methanol cost equal to 2.4 €/kg).**



Figure 7 shows the cost breakdown for an average cost scenario. One can see that the main cost sources are the financing cost (33% of total methanol cost), the energy ship's capital cost (hull + Flettner rotors + water turbines + auxiliaries and integration, 17% of total methanol cost), and operation and maintenance cost of the FARWINDERs (16%). The total cost of energy storage - including the power-to-methanol plants capital cost and maintenance cost, CO2 supply, and tankers capital cost and operation and maintenance cost - accounts for 25% of total cost.

## 4.2 Long-term cost and market potential

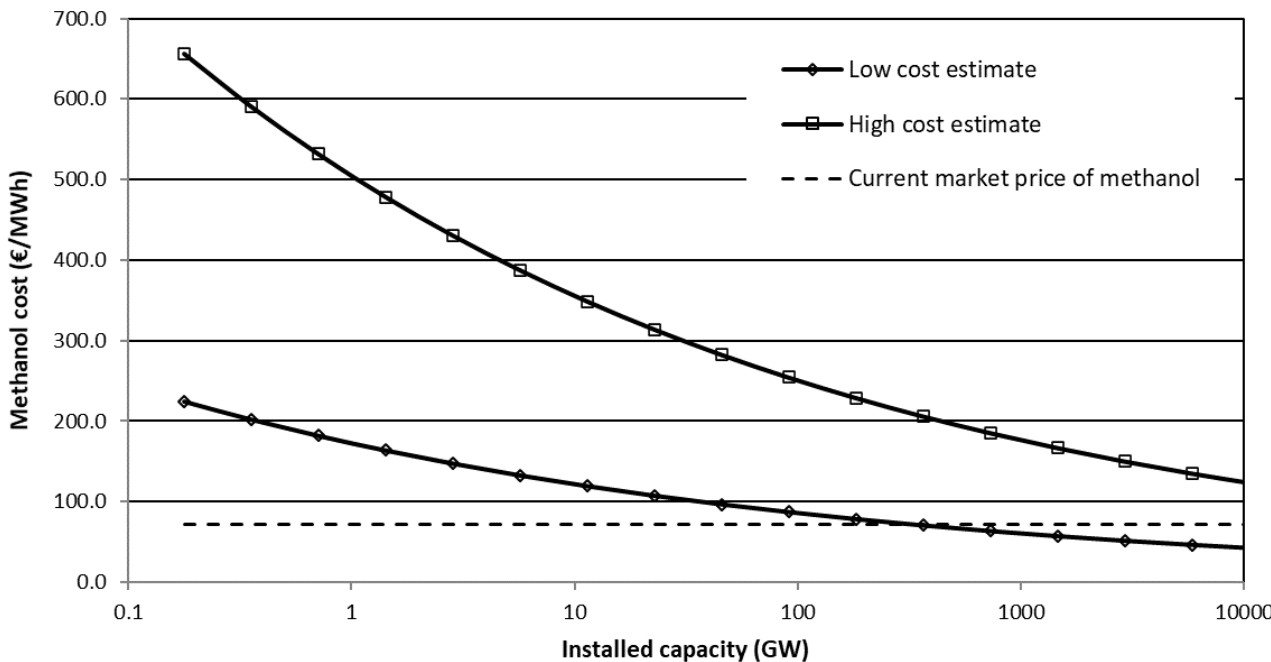

**Figure 8  Cost of methanol produced by FARWIND systems as function of the installed capacity and comparison with the current market price of methanol produced from fossil fuels and fossil feedstocks.**

However, as for the energy ships, one can expect that the cost of FARWIND systems will decrease with increasing installed capacity. Fig. 8 shows the expected cost reduction for the methanol cost as function of the installed capacity. A learning rate of 10% was assumed (as for the energy ships, see section 4.2). One can see that it would take thousands of GW of installed capacity to achieve competitiveness with methanol produced from fossil fuels.

Let us now consider the perspective of FARWIND-produced fuel for the transportation fuel market. Indeed, methanol can be blended with gasoline in low quantities for use in existing road vehicles. According to (Methanol Institute, 2014), the blend can include up to 15% methanol by volume (M15 fuel). Moreover, flexible fuel vehicles which can run on an 85%–15% methanol–gasoline mix (M85 fuel) have been developed and commercialized (e.g. the 1996 Ford Taurus); and M100





(100% methanol) vehicles are in development (Olah et al., 2018). Thus, FARWIND-produced methanol could be used as a low-carbon substitute to oil on the transportation fuel market.

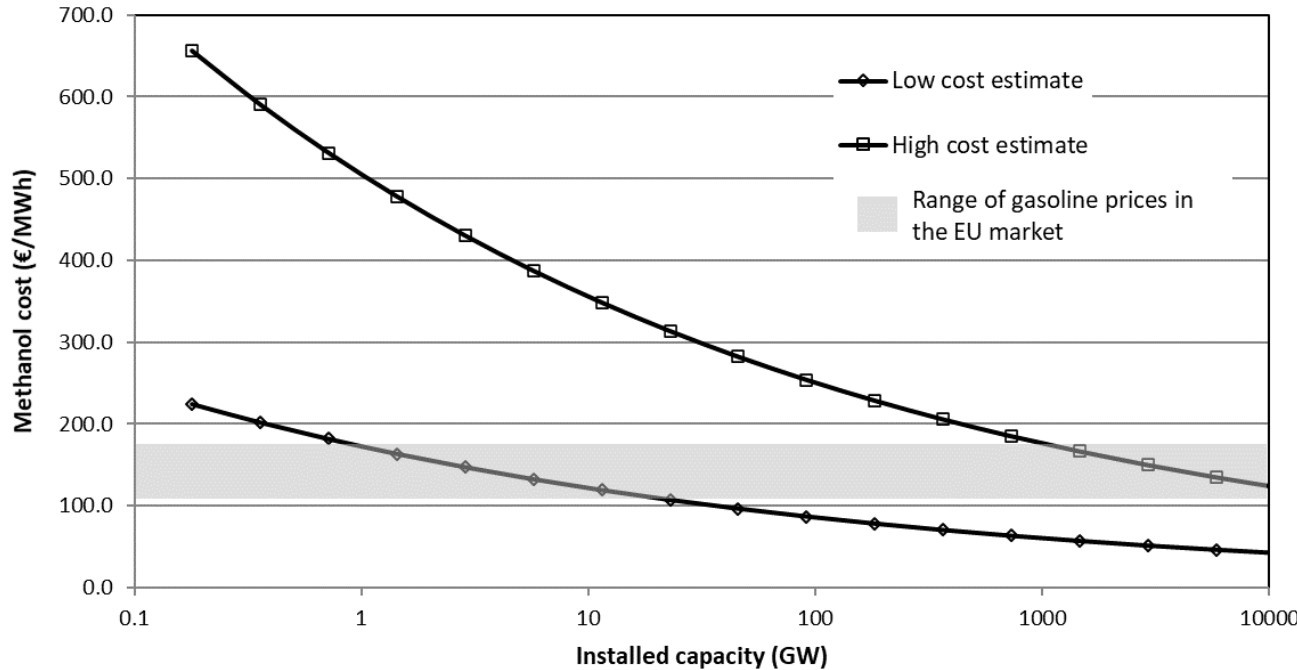


**Figure 9 Cost of methanol produced by FARWIND systems as function of the installed capacity and comparison with current market price of gasoline in the EU**

Let us compare the cost of FARWIND-produced methanol to the gasoline price in the EU. Gasoline price ranges from 1.1 €/L (Bulgaria) to 1.7 €/L (Netherlands), the price differences arising from different policies on fuel taxes in different
countries (European Commission, 2019). This price range is equivalent to 112 to 173 €/MWh$_{th}$, since the standard density of gasoline traded in the EU is 0.755 kg/L and its energy content is approximately 13 kWh$_{th}$/kg. Thus, as can be seen in Figure 9 and provided that taxes policies are favourable to FARWIND-produced methanol, it may take "only" a few tens of GW of installed capacity to be competitive with gasoline on the EU transportation fuel market.

**4.3 Comparison with methanol production by offshore wind turbines**

Finally, let us compare the cost of methanol production by FARWIND systems and offshore wind turbines. In this respect, we assume that the first-of-a-kind FARWIND system is deployed by 2030. At that time, according to (IRENA, 2019), the global offshore wind energy capacity will have reached 230 GW.

The key economic drivers in power-to-gas or power-to-liquid processes are the cost of input electricity to the power-to-gas/liquid plant and the power-to-gas/liquid plant capacity factor (Fasihi et al., 2016; Ioannou and Brennan, 2019).
Based on that data, one can calculate the methanol production cost using:





$$LCOM = \frac{(CRF + \lambda')I'}{8760 \times C_F \times P_{rated} \times \eta_{MeOH}} + \frac{LCOE_{elec}}{\eta_{MeOH}} + 1.38 \times \frac{C_{CO2}}{LHV_{MeOH}}$$

(2)

where $I'$ is the capital cost of the power-to-methanol plant, $\lambda'$ is the O&M rate of the power-to-methanol plant plus the insurance rate, $C_F$ is the plant capacity factor, $P_{rated}$ is the rated power of the plant, $\eta_{MeOH}$ is the plant efficiency (49%, see

(Babarit et al., 2020)), $LCOE_{elec}$ is cost of input electricity to the power-to-methanol plant, $C_{CO2}$ is the CO$_2$ cost per unit mass and $LHV_{MeOH}$ is the lower-heating-value of methanol per unit mass (the factor 1.38 corresponds to the fact that it takes 1.38 kg of CO$_2$ to produce 1 kg of methanol).

| CAPEX | 480 – 1,285 €/kW |
|---|---|
| OPEX | 6 – 7% |
| CO$_2$ supply cost | 20 – 200 €/t |
| Insurance | 0.6 – 1.2% |
| Lifetime | 20 – 25 y |
| WACC | 6 – 10% |

**Table 4 Expected costs of a power-to-methanol plant by 2030 (excluding input electricity)**

Table 4 shows the cost assumption for the power-to-methanol plant of the offshore wind farm. The capital cost is

assumed to be a third of that of the first-of-a-kind FARWIND system as the power-to-methanol plant would be much larger (Brynolf et al. 2018) and as it may be shore-based. According to (IRENA, 2019), the cost of electricity from offshore wind farms will be in the range 40 to 80 €/MWh by 2030 with capacity factors in the range 36 to 58%. Therefore, using Eq. 2, we find that the methanol production cost by offshore wind farms would be in the range 110 to 375 €/MWh$_{th}$ (0.6 to 2.1 €/kg) by 2030. Thus, by 2030, the cost of methanol produced by a FARWIND energy system (1.3 to 2.1 €/kg) would not be

competitive with that produced by a shore-based power-to-methanol plant powered by a large offshore wind farm.

However, that would be the case for a first of a kind for FARWIND, whereas it would be for an expected global capacity of 230 GW for offshore wind turbines. Therefore, provided that sufficient FARWIND capacity is installed, FARWIND-produced methanol may become comparable to that of offshore wind farms-produced methanol. This is shown in Figure 10 which shows a comparison of the long-term methanol cost produced by FARWIND systems and by offshore

wind farms. A learning rate of 10% was assumed both for the FARWIND systems and for the methanol-producing offshore wind farms. However, for the offshore wind farm, it has been taken into account that the cost of input electricity assumes an installed 230 GW global offshore wind capacity. Therefore, it can be expected that it would take a further 230 GW to achieve a cost reduction of 10% of that part of the methanol cost (second term in Eq. 2.). Thus, the methanol production cost of offshore wind farms as function of the installed capacity $C_{OW}$ (in GW) can be written:

$$LCOM_{OW}(C_{OW}) = \left( \frac{(CRF + \lambda')I'}{8760 \times C_F \times P_{rated} \times \eta_{MeOH}} + 1.38 \times \frac{C_{CO2}}{LHV_{MeOH}} \right) \times 0.9^{\log_2 \frac{C_{OW}}{0.2}} + \frac{LCOE_{elec}}{\eta_{MeOH}} \times 0.9^{\log_2 \frac{230+C_{OW}}{230}}$$

(2)



Note that, in agreement with the cost data of (Brynolf et al., 2018), it has been assumed that the capacity of the first methanol-producing offshore wind farm is 200 MW.

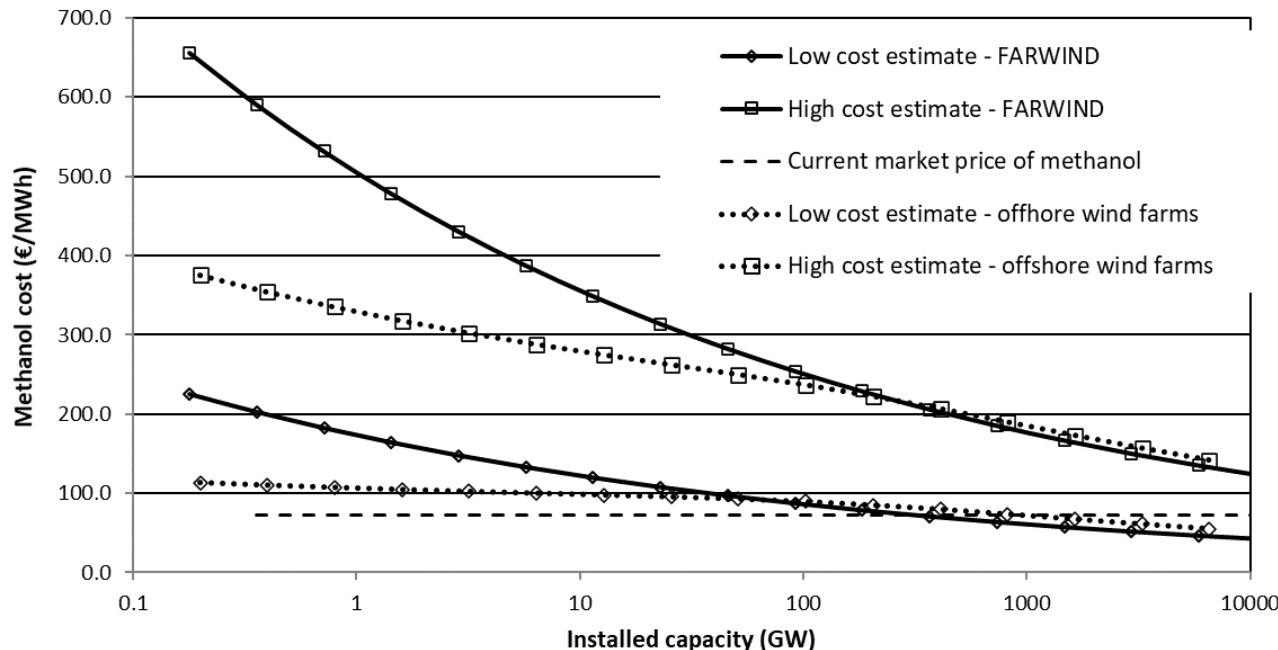


**Figure 10**        **Comparison of long term methanol cost produced by FARWIND systems and offshore wind farms as function of the installed capacity**

## 5 Conclusions

In this paper, we proposed an energy system for sustainable methanol production from the far-offshore wind energy

resource. It is based on an autonomous fleet of 112 energy ships and 5 manned tankers for the collection and transport of the produced methanol, as well as the supply of $CO_2$ to the energy ships. Its methanol production is expected to be in the order of 70,600 t per annum (approximately 390 GWh per annum of chemical energy). The cost of this methanol is expected to be in the range 1.2–3.6 €/kg for the first-of-a-kind FARWIND system, which is significantly greater than the current market price for fossil fuel-derived methanol (0.4 €/kg). However, methanol can be used as a substitute to fossil fuels on the fuel

transportation market: since the price of transportation fuel is high in most European countries, and assuming that a cost reduction similar to that observed for land-based wind energy can be achieved, the cost of FARWIND-produced methanol could compete with gasoline in the EU.

The cost of methanol produced by a first-of-a-kind FARWIND system is unlikely to be competitive with that produced by a large shore-based power-to-methanol plant powered by an offshore wind farm. However, provided that



sufficient FARWIND capacity is installed, FARWIND-produced methanol may become comparable to that of offshore wind farms-produced methanol. Moreover, one should note that the cost of FARWIND-produced methanol is based on a particular energy ship design, which might be optimized to reduce costs.

## 6 Acknowledgements

This research was partially carried out in the Marine Energy Alliance project, which is financially supported by Interreg
North West Europe.

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
