# Peer review of "Exploitation of the far-offshore wind energy resource by fleets of energy ships - Part 2: Updated ship design and cost of energy estimate"

_Wind Energy Science, 2021_

## Author Response (AR1)

Dear anonymous referees,

Thank you very much for reviewing our manuscript. We greatly appreciate your comments and suggestions. We have revised the manuscript accordingly. Please find below our point-by-point responses to your suggestions and concerns.

**Reviewer #1**

**General comments**

This article documents the early design stages of a creative, renewable Power-to-Liquid system. The article is written in generally good language. The technical aspects of the proposed energy ship are documented appropriately for a case study and assumptions/references are transparent. The economic evaluation of the concept is documented transparently too.

No answer required.

The main critique point is specific comment no. (10). The critique refers to the methanol price projections and is decisive for the market potential of the proposed solution and eventually the conclusion of this article. I recommend this point being double-checked by another reviewer.

Please find below our detailed answer to your comment no (10).

**Specific comments**

1. Line 22: "the cost may be comparable to that of methanol produced by offshore wind farms in the long term" – see specific comment no. (10).

   See our detailed answer to your comment no (10).

2. Line 35: It would be helpful for the reader if you shortly mentioned up to three main reasons for your choice of methanol, based on your referenced previous assessment.

   The following text has been added in the introduction.

   *In the proposed system, the fuel is methanol. Hydrogen was not retained because it was found in Babarit et al. (2018) that hydrogen storage and transportation costs could account for nearly half of the cost of the delivered hydrogen when it is produced far-offshore (because of the low volumetric energy density at ambient temperature and pressure conditions which is a well-known challenge for hydrogen storage and transportation). In contrast, the other possible energy vector options (synthetic natural gas (SNG), methanol, or Fischer–Tropsch fuel (FT fuel), Graves et al., 2011; and ammonia, Morgan, 2013) are much simpler to store, transport and distribute (particularly methanol and FT fuel, as they are liquid for standard conditions of temperature and pressure). Moreover, they can be incorporated into existing infrastructure with little to no modification. The drawback is that they each require the supply of an additional feedstock (carbon dioxide or nitrogen depending on the energy vector) and an additional conversion step in the energy conversion process. The additional conversion step decreases the overall energy efficiency and increases the size and complexity of the PtX plant. In a previous study (Babarit et al., 2019), we investigated whether these drawbacks could be compensated for by the easier storage, transportation and distribution of the products, and we found that methanol is the most promising solution; hence it is retained as the energy vector in this study.*

3. Line 55 and following: As far as I understand, your proposed design has progressed and you provide comparisons/updates to previous estimates. This documentation in itself may be of value, as it showcases how weight or cost estimates develop throughout subsequent design stages. A short sentence highlighting this value could bring attention to this aspect.

Thank you for the suggestion. The text after line has been modified accordingly :

*The overall aim of the present study is to investigate the energy and economic performance of the FARWIND energy system. A preliminary energy ship design was proposed in (Babarit et al., 2020) and its energy performance was investigated. The cost of energy was estimated in (Babarit et al., 2020b). It was found that an initial FARWIND system could produce approximately 100,000 tonnes of methanol per at a cost in the range 0.9 to 2.1 €/kg.*
*This preliminary design has been reviewed by ocean engineering and marine renewable energy's experts of the Marine Energy Alliance European project (EMEC, 2020); and wind-assisted propulsion experts (Blue WASP, 2020). Based on their feedback, the ship design has been progressed; and an the economic model has been refined. The aim of the present paper is to present that improved design, the economic model, and the resulting cost of energy. The present study also provides an example of how cost estimates develop throughout subsequent design stages.*

4. Lines 67 & 88: you refer to eq. 2 from Babarit et al. 2020 twice, hence it seems to be relevant for this study. Consider showing that equation explicitly here instead of only referring to the previous article

That equation has been added :

*The propulsive force (thrust) T of a Flettner rotor depends on the lift coefficient CL, the drag coefficient CD,the apparent wind speed V, the apparent wind angle α, the rotor area A (height times diameter) and the air density $\rho_a$:*

$$T = \frac{1}{2}\rho_a A V^2 (C_L \sin\alpha - C_D \cos\alpha)$$

*(1)*

5. Line 82 (Figure 4): You could indicate the vector of the propulsive force with an arrow in the left part of the figure. Potentially four arrows with lengths proportional to each FR's force contribution.

The vectors of propulsive force are now indicated in Figure 4. Thank for the suggestion.

6. Line 97: since the displacement has changed, I assume the hull shape has changed too. 'The hull shape (Wigley hull) has been updated based on a more accurate displacement estimate' could clarify this.

Actually, the hull shape has not changed. It is only the draft that has changed. It was 1.6 m in the initial design. It is 2.1 m in the new design. We forgot to update Table 1 in the initial submission. This mistake is now corrected. To clarify, the following sentence has been added in section 2.2:

*The draught has increased from 1.6 m for the initial design to 2.1 m for the updated design.*

7. Lines 116-122: Consider mentioning the efficiency of the H2-to-methanol plant as well in order to increase transparency.

   The efficiency of the hydrogen-to-methanol plant has been added explicity in section 2.4:

   *Assuming the same 60% efficiency for the electrolyzer and the same 78% efficiency for the hydrogen-to-methanol plant as for the initial design, the rated power of the hydrogen-to-methanol plant is 680 kW (850 kW for the initial design).*

8. Lines 182 & 201: You could improve understanding by framing the annual methanol production capacity in terms of vehicles powered. E.g. units of 5000 dwt bulk carriers propelled:
   70,600t/year = 388,300MWh/year chemical energy assumptions annual energy consumption

   bulk carrier: 1,410kW x 24h/day x 180days/year = 6,091MWh/year

   6,091MWh / 50% thermal engine efficiency = 12,182 MWh/year chemical energy

   388,300MWh / 12,182MWh = 32 vessels that could be powered by the designed fleet

   *For sake of illustration, let us estimate the number of 5,000 t bulk carriers which could be powered by a FARWIND system. As mentioned in section 3.1, their propulsion power is 1,410 kW for a service speed of 12 knts. Assuming that they would sail at that speed 292 days per year (80% of the time) and that their engine efficiency is 40%, the required chemical energy is approximately 24,700 MWh per year. 70,600 t of methanol corresponding to approximately 386,000 MWh of chemical energy, the designed FARWIND energy system could power approximately 16 5,000 t cargo vessels.*

9. Section 4.2 and 4.3: Would it be more logical to switch the order of these two sections?

   A comparison of alternative carbon-neutral methanol production pathways first and market potential second (potentially only of the best candidate solution) seems more intuitive.

   Agree. The order of the sections has been changed and the text has been updated accordingly.

10. Figures 8, 9 and 10 and lines 360-364: If I understand the concept of learning rate correctly, you assume that the (levelized) cost of methanol decreases by 10% for each doubling in capacity. Many of the capital-intensive systems (shown in Figure 7) use existing technologies, and in particular technologies that are used in offshore windfarms and connected methanol production plants too. The cost for the same technology however will not develop significantly differently depending on whether the technology is installed onboard the energy ship or in offshore wind farms. Put differently, the cost decrease should be seen in relation to the worldwide installed capacity of the technology, not the energy ship (or fleet) alone. In that case, the costs of the energy ship would not fall as quickly as projected and the system thus not be competitive.

    On the other hand, it may be argued that the cost of offshore wind methanol increases with increasing installed capacity, as windfarms need to move to more distant offshore locations.

The energy ship seems to be a rather robust solution to this issue, as it is relatively insensitive to shore distance and water depths.

I recommend these cost projections being carefully double-checked. They do not affect the technical assessment, but have a significant effect on the market potential and hence the conclusion of this article.

We agree that the conclusion heavily depends on the assumption for the learning rate.

Your point is that a 10% learning rate is too optimistic because most technologies used in the FARWIND system are existing. Our point of view is that it is actually on the conservative side, for the following reasons.

First, the number of technologies which are truly existing and fully established is actually limited, and/or for existing technologies they are not mass-produced:

- The rotors account for 30 to 45% of the energy ship's CAPEX. The number of rotors which have been installed to date is no more than 20.
- The water turbine is new (14 to 16% of the cost). There are no water turbines available on the market which match the requirements of the energy ship (MW rated power and 10 m/s flow velocity).
- Regarding the hull (12% to 13% of the cost) and the tanker (8% of the cost), despite shipbuilding is an old industry, there is a 10% series effect on the workload according to the OECD (https://www.oecd.org/industry/ind/37655301.pdf, page 8). This is not really surprising as most of the time a new ship is also a new design.
- Regarding assembly and integration (12 to 14%), this cost can be expected to reduce significantly with the development of dedicated tooling.
- The electrolyzer account for 12 to 13% of the cost. To date, there are approximately 200 MW of installed electrolyzer capacity. Thus, this cost can be expected to reduce very significantly with development of the electrolyzer industry (GWs of deployments have been announced).
- H2-to-MeOH plant (4 to 5%): approximately 90 GW of methanol production capacity are operating to date. Nevertheless, the level of standardization can be expected to be very low as every production site is different. Therefore, similar cost reductions as for the hull could be achieved.

Second, and the most important, there is a significant difference between the learning rate of installed cost (CAPEX/kW) and the learning rate of levelized cost of energy (LCOE, in €/MWh). According to [IRENA, 2021][1], the learning rate of offshore wind installed capacity for the period 2010 to 2020 has been 9.4%. However, the LCOE learning rate has been 15%. For onshore wind, over the same period, the installed cost learning rate has been 16.6% while the LCOE learning rate has been 32%.

In our study, we considered a 10% learning rate both for the installed cost and LCOE cost, which is thus very conservative in comparison to what has been observed over the last ten years.

**Technical comments**
* * *
[1] https://www.irena.org/publications/2021/Jun/Renewable-Power-Costs-in-2020, page 37 to 39

1. Line 16: consider taking out the reference from the abstract.

   ✓ Taken out

2. Lines 16-17: you mention the "energy performance has been assessed". Hence the statement "aim is to estimate the energy […] performance" seems confusing. 'Revisit' or 'update based on design progression' might clarify this.

   ✓ Modified, thanks for the suggestion

3. Line 18: "wind-assisted propulsion experts" (without 's)

   ✓ Corrected, thanks.

4. Line 30: consider replacing "low-carbon alternatives" by 'climate-neutral'/'carbon-neutral' or similar.

   ✓ Replaced by carbon-neutral.

5. Line 32: '**a** sustainable fuel' or 'sustainable fuel**s**'

   ✓ Corrected, thanks.

6. Lines 38-39: consider replacing "sustainable" by 'carbon/climate-neutral' or similar to be more precise.

   ✓ Replaced by carbon-neutral.

7. Line 49: Do you mean 'levelized' cost of energy? In that case, it can be advantageous to mention that explicitly.

   ✓ Yes, "levelized" has been added.

8. Line 58: Figure 2 (not 3)?

   ✓ Yes, Figure 2.

9. Line 61: Consider replacing "Justifications" by 'explanations' or similar.

   ✓ Replaced by "explanations", thanks.

10. Table 1: Be consistent with using either $H_2$ or H2 and CO2 or $CO_2$

    ✓ Corrected.

11. Line 71: 'formulas' or 'a formula'

    ✓ Corrected, thanks.

12. Line 230: Consider making an ordinary reference to this weblink.

✓ The link has been put in the references.

13. Figure 7: an exploded pie chart (pieces grouped by CAPEX, OPEX and others) can improve the understanding of the figure.

✓ As we do not know how to group pieces in an exploded pie chart in excel, we used colours to make groups.

14. Line 401: The title of this reference seems to be wrong.

✓ Corrected.

**Reviewer #2**

**Summary**

The present paper examines a novel wind-to-liquid power conversion system (energy ship) and an energy infrastructure (FARWIND) with respect to energetic and economic performance.

The FARWIND system comprises a fleet of energy ships that harvest wind energy far-shore and convert in on-board to methanol, a smaller fleet of tankers that provide feedstock and collect produce, and on-shore terminals.

Energy ships are sailing ships with water-turbines attached at the hull to provide energy to a power-to-X process. In the present paper, methanol was chosen for energy storage. Tankers firstly provide cryogenic $CO_2$ that is used in the power-to-methanol process on-board the "energy ships" and secondly collect produced methanol which is then discharged at the on-shore terminals.

A previously developed model and preliminary design form the basis of the analysis. In the first part of this contribution, the technical model and preliminary design are revised. In the second part, an economic feasibility study for the FARWIND system is carried out.

No answer required.

**technical model revision**

The authors present a revision to a preliminary design presented in an earlier contribution to "Wind Energy Science". The design features Flettner rotors for propulsion, a catamaran hull, and two turbines attached on either side of the hull. Revisions to the design include height of the rotors and rated power of the turbines. Model revisions include

* improved formulae to estimate aerodynamic coefficients of the rotors based on empirical data at higher (more realistic) Reynolds number,

* consideration of the effect of spin ratio on rotor driving power,

* consideration of rotor-rotor interaction,

* consideration of atmospheric boundary layer,

* revised mass-scaling of the hull, resulting in twice the mass of their preliminary design, and

* a revised turbine mass-estimate based on expert advice.

As a result, the authors report 10 to 20% less power generated than initially predicted.

No answer required.

**economic model**

Assumptions on service-cycle length and annual production rates are made including the power predictions from the technical model. The following analysis is formed on the basis of one tanker servicing 28 energy ships per week for 4 weeks until returning to a terminal at the shore.

Tanker weight and corresponding propulsion power are estimated from service time and required tank volume.

The authors estimate an annual methanol production of approx. 70 600 t/a if continuous production is to be ensured, while factoring in production downtime due to failures and maintenance.

CAPEX for individual components including cost reduction for the entire FARWIND system due to scale effects are estimated based on literature research. Expected maintenance and operation as well as insurance costs are assumed to be proportional to capital costs. Expected ranges are taken from literature, except in the case of hull auxiliary and tanks which are arbitrarily assumed to be 2%!

To assess economic performance, levelized cost of methanol are computed under uncertainty, yielding a range of 1.2 to 3.6 Euro per kilogram, which is reported to be three times higher than usual market prices.

With respect to model assumption and uncertainty, it is found that:

1.  Even at a learning rate of 10% (scale effect) the FARWIND system would not be profitable for reasonable installed capacity at current market prices for methanol.
2.  If the produced methanol was used as an alternative fuel source, prices could be competitive with current gasoline prices in the European Union.
3.  When benchmarked against a hypothetical power-to-methanol wind farm, the FARWIND system is may become competitive long term for large installed capacity.

No answer required.

**General remarks**

As is revealed in figure 6, the previous assumption on required power to drive the rotors (4 x 40kW = 160kW) deviates significantly from the new model for a number of TWS and TWA combinations! Similarly, predictions for generated power reduced significantly as a result of model improvement. This hints at the fact that it might be advisable to investigate other parts of the technical model for further possibilities of improvement. Even though part one of the article is seen as an update to previous work, the discussion of the energetic performance model is kept too brief, as it leaves a few open questions. For example, power generation is surprisingly steady for different TWA and const. TWS while the peak-power stagnates with increasing TWS, which seems counter intuitive at first. I suggest that either, behaviour of the system at different TWS and TWA should be discussed in more detail, a reference to such discussion is given, or reports on model revisions should be shortened to shift the focus of the analysis.

Your first general comment is about the fact that in Figure 6, one can see that the generated power of the energy ship is steady for different true wind angles and constant true wind speed while the generated power stagnates with increasing true wind speed. This is because the Flettner rotors rotational speed and the water turbines' induction factor are controlled in order to maximize power production while satisfying the constraints (maximum rotation velocity and thrust force for the rotors, maximum power generation for the water turbine). In the proposed design, the maximum power generation of the water turbines is limited to 1,600 kW (2 x 800 kW). Similar to wind turbines, there exists a rated wind speed above which the available wind energy exceeds the conversion capability of the energy ship (in other words, the ship could produce more power if it is equipped with generators of greater capacity). One can see in Figure 6 that for the proposed design, this rated wind speed is approximately 10 m/s. For wind speeds above that threshold, the rotational velocity of the Flettner

rotors reduces (corresponding to the pitching of the blades for a wind turbine) in order to maintain maximum power generation while reducing the rotors' power consumption (panel c) in Figure 6). To clarify, the following sentence has been added after Figure 6:

*"Note that for each data point, the water turbine's induction factor and the rotors' spin ratio were optimized in order to maximize power production while satisfying the constraints (maximum rotation velocity and thrust force for the rotors, maximum power generation for the water turbine)."*

In general, the analysis is based on many broad assumptions that undoubtably include considerable uncertainty. The notion of uncertainty is addressed by considering ranges for most parameters. There is however, no mention of distributions within the identified ranges. The expected rate of production, on the other hand, is assumed without any notion of uncertainty. It remains unclear how uncertainty is propagated through the model! It should be clarified which method of error propagation was used. For example, without the notion of distributions, it remains unanswered if, based on the assumptions, it is equally as likely to yield lower or upper LCOM as reported in figures 8 to 10. Besides, the mean with error bars would arguably more appropriate presentation in this context.

Your second general comment is related to uncertainty, and more specifically to (i) the fact that no distributions are provided for the identified uncertainty ranges, and (ii) the fact that the rate of production does not include uncertainty.

Regarding (i), the uncertainty ranges are based on suppliers, experts' recommendations and/or publicly available literature. Unfortunately, none of these sources provided distributions. Regarding (ii), we believe that it would be arbitrary to put an uncertainty on a number which is the result of a numerical model. Comparisons with experiments (which are not yet available) would be necessary to determine the level of accuracy.

Regarding the propagation of uncertainty, the low end of the LCOM (respectively high end) was obtained by using the most optimistic cost data (respectively most pessimistic cost data). Therefore, it is equally as likely to yield lower or upper LCOM. Therefore, we modified Figures 8 and 10 (now Figures 8 and 9) following your recommendation to show the mean and error bars. The following sentence has also been added at the end of the first paragraph of section 4.1:.

*"Note that the low end of the range (respectively high end) was obtained by using the most optimistic cost data (respectively most pessimistic cost data)."*

For a preliminary case study, the method of determining economic feasibility is probably sufficient. As the analysis was based on a predetermined design, the validity of the results is at the current state questionable. If the design of the "energy ship" was optimized for the specific purpose of increased profitability, the proposed system might become significantly more competitive compared to the current design, as noted in section 5.

I suggest to accept with minor revisions (see below)!

 **# Revisions**

* Section 2: behaviour of the system at different TWS and TWA should be discussed in more detail, a or a reference to such discussion should be given, or reports on model revisions should be shortened to shift the focus of the analysis --> see general remarks

See our answer to your general remarks.

* line 71: no definition of the Reynolds number is given

The definition of the Reynolds number has been added in the revision of the paper: *"(…), with the Reynolds number defined as: $Re = \frac{VD}{v}$ (2) where v is the kinematic viscosity and D is the rotor diameter."*

* Sections 2.4 to 2.6 list assumptions for the power-to-methanol plant, tanks and auxiliary equipment: No references were given! They might be included in the first part, but this isn't stated either. References are given later in section 4.1, it's unclear however, if those are the ones considered in 2.4 to 2.6 as well.

Indeed, the references are included in the first part. It is clarified in the revision of the paper:

*"2.4 Power-to-methanol plant*

*(…)*

*Assuming the same 60% efficiency for the electrolyzer and the same 78% efficiency for the hydrogen-to-methanol plant as for the initial design (Babarit et al., 2020), the rated power of the hydrogen-to-methanol plant is 680 kW (850 kW for the initial design). Its weight estimate is 17 t (24 t for the initial design).*

*2.5 Storage tanks*

*The capacities of the storage tanks ($CO_2$ and methanol) are set such as they can accommodate 7 days of production at rated power (approx. 17 t of methanol). Thus, the $CO_2$ tank weight is 15 t and that of the methanol tank is 4 t (Babarit et al., 2020),.*

*2.6 Auxiliary equipment*

*As for the initial design (Babarit et al., 2020),, the weight of the auxiliary subsystems is taken equal to 10% of the total mass budget excluding the hull weight (41 t)."*

* Figure 6: polar plots are missing units of measure for power and speed!

Yes indeed. This mistake is corrected in the revision of the paper.

* line 301: Please double check the units! The market price of methanol is given as 0.4 Euro per kilogram or 72 Euro per Megawatt hour. With carbon tax it is given as 6 or 13 Euro per Megawatt hour depending on the taxation, which is about ten times lower than the price given w/o tax.

You may have read this paragraph too quickcly. 6 €/MWh to 13 €/MWh is not the price with taxation but the price increase with taxation: *"In 2018, the carbon tax was 44.6 €/ton in France and 110 €/ton in Sweden; if $CO_2$ emissions were taken into account, the methanol price would increase by 6 €/MWhth and 13 €/MWhth respectively."*

* line 401. The title of the reference seems to have changed. Consider adding DOIs to your references where possible!

Yes, there was a mistake in the title (and list of authors) of this reference. It has been corrected. The DOIs have also been added wherever possible.

---

## Author Response (AR2)

**Editor's comment:**

Thank you for improving this paper which I know has been a long way in the process. Please do include the extended uncertainty discussion as suggested by the second reviewer and the associate editor. Sincerely, Jakob Mann

Dear Editor,

In order to address this issue, we added a new section at the end of the manuscript and we modified the end of the conclusion :

*4.4 Uncertainty discussion of cost of energy*

*In the present study, the economic feasibility is based on broad assumptions that undoubtably include considerable uncertainty. That uncertainty has been taken into account by considering ranges for the cost parameters. The ranges were determined based on suppliers and/or experts' recommendations, and/or publicly available literature. The uncertainty was propagated by applying the most optimistic cost data (respectively most pessimistic cost data) to determine the low end (respectively high end) of the levelized cost of methanol.*

*Regarding energy production, no uncertainty was applied. This is because energy production results from a deterministic numerical model. Comparisons with experiments (which are not yet available) are necessary to determine its level of accuracy. This may lead to the higher end of the cost of methanol estimate actually be underestimated should the actual energy production be significantly smaller than that predicted by the numerical model. On the other hand, if the design of the energy ship was optimized for the specific purpose of increased profitability, the proposed system might become significantly more competitive compared to the current design.*

*5. Conclusion*

*(...)*

*However, one should note that the present study is based on many broad assumptions that include considerable uncertainty. Further work is needed to confirm the findings. Moreover, one should note that the cost of FARWIND-produced methanol is based on a particular energy ship design, which might be optimized to reduce costs.*

We hope that it corresponds to your request.

Best regards,

Aurélien Babarit